# Validity and Reliability of the Turkish Version of the Screening Questionnaire of Highly Processed Food Consumption (sQ-HPF)

**DOI:** 10.3390/nu16152552

**Published:** 2024-08-03

**Authors:** Nazlıcan Erdoğan Gövez, Eda Köksal, Celia Martinez-Perez, Lidia Daimiel

**Affiliations:** 1Department of Nutrition and Dietetics, Faculty of Health Sciences, Gazi University, Ankara 06490, Turkey; betkoksal@yahoo.com; 2CIBER Fisiopatología de la Obesidad y Nutrición (CIBEROBN), Instituto de Salud Carlos III (ISCIII), 28029 Madrid, Spain; celia9124@gmail.com; 3Nutritional Control of the Epigenome Group, Precision Nutrition and Obesity Program, IMDEA Food, CEI UAM + CSIC, 28049 Madrid, Spain

**Keywords:** highly processed food, ultra-processed food, validation, validity and reliability, adaptation, nutrition, non-communicable diseases

## Abstract

The global consumption of highly (ultra) processed foods (HPFs) is increasing, and it is associated with non-communicable diseases. This study aimed to assess the validity and reliability of the Screening Questionnaire of Highly Processed Food Consumption (sQ-HPF). This study included 94 adults. Sociodemographic data were collected, and anthropometric and blood pressure measurements were performed. The sQ-HPF was translated into Turkish and culturally adapted. Dietary intake was assessed using three-day dietary records. Factor analysis and Cronbach’s alpha were used to evaluate the validity and consistency of the sQ-HPF. Test–retest reliability was assessed with the intraclass correlation coefficient (ICC). Three items from the original sQ-HPF were excluded due to low factor loadings. The Kaiser–Meyer Olkin (KMO) coefficient for the measure of sample adequacy was found to be 0.642 and Bartlett’s test of sphericity was found to be significant (*p* < 0.001). A significant correlation was found between the sQ-HPF score and HPF consumption derived from the 3-day dietary records (*p* < 0.05). Cronbach’s alpha was found to be 0.65. Individuals with higher sQ-HPF scores consumed a significantly greater percentage of energy from HPFs (kcal/day) (*p* < 0.001). The sQ-HPF demonstrated good test–retest reliability (ICC = 0.76). The Turkish version of the sQ-HPF is a valid and reliable tool for assessing HPF consumption patterns and can be used in epidemiological and clinical studies.

## 1. Introduction

Non-communicable diseases (NCDs) account for 74% of all fatalities worldwide, or the demise of 41 million individuals annually. Each year, 17 million individuals under the age of 70 die from NCDs and low and middle-income countries account for 86% of these premature fatalities. Annually, cardiovascular diseases are responsible for the death of 17.9 million individuals due to NCDs; moreover, malignancies follow with 9.3 million, chronic respiratory diseases with 4.1 million, and diabetes with 2.0 million, including kidney disease-related fatalities caused by diabetes [1]. Globally, the burden of NCDs has steadily risen in recent years; in 2021, NCDs accounted for 85 percent of years lived with disability and 90 percent of fatalities in the World Health Organization (WHO) European Region [2]. Tobacco use, physical inactivity, alcohol consumption, and an unhealthy diet are the four primary behavioral factors associated with mortality from NCDs [3].

The nutrition transition model is a concept proposed to describe the major changes in human nutrition and activity patterns that have occurred from the 1990s to the present [4]. Advertising, marketing, and technological advancements that influence the distribution network of foods have all played a role in the nutrition transition, in addition to fundamental factors such as income level, price changes, and urbanization that have an immediate impact on nutrition and physical activity patterns [5]. There has been a global shift toward diets that are low in fiber and micronutrients and high in saturated fat, sodium, sugar, and refined carbohydrates. The term “Western diet” has been attributed to this phenomenon due to its prevalence, which is particularly notable in the United States, the United Kingdom, and European nations [6].

Variations in food processing techniques have been observed since ancient times [7]. This trend has continued throughout history, particularly since the Industrial Revolution and economic globalization, which commenced with the rise of humanity [8]. Food processing is crucial in ensuring the production of consumable, safe, and nutritious food for the community, as well as in maintaining its longevity. Processing may prolong the shelf life of foods by inhibiting the growth of microorganisms, thereby lowering the risk of food-borne illnesses. On the other hand, processing may also enhance the digestibility of foods [9].

Foods are classified differently based on the level of processing and may be classified depending on the extent and objective of processing. This classification considers all the physical, biological, and chemical techniques employed in the food manufacturing process, including the utilization of additives [8]. The commonly utilized classifications for food processing include NOVA [10], the National Institute of Public Health of Mexico (NIPH) [11], the International Agency for Research on Cancer–The European Prospective Investigation into Cancer and Nutrition (IARC-EPIC) [12], and International Food Information Council (IFIC) [13] classifications. Irrespective of the classification method used, the global production and consumption of processed food is steadily rising, making it crucial to evaluate its impact on health.

Highly (ultra) processed foods (HPFs) have emerged as a substantial and, in certain instances, the primary source of dietary energy in high-income countries, such as the United States, Canada, the United Kingdom, and Australia, since the 1950s onwards [14]. These cuisines have only become genuinely accessible on an international level in the present era, which is marked by the globalization of food systems [8]. At present, HPFs hold a significant position in the nutrition transition that is taking place in low and middle-income countries; the nutrition transition entails a departure from conventional diets and a shift towards those that are associated with obesity and diet-related NCDs [15].

The presence of numerous classification systems for food processing, each using distinct criteria, can result in disparate conclusions concerning health outcomes [16,17]. A considerable number of studies determine HPF consumption through the assessment of dietary records, 24 h recalls, or food-frequency questionnaires (FFQs) [18,19,20,21,22]. However, determining consumption through these methods creates difficulties in standardizing the results and is time-consuming. HPF consumption has been determined by these methods or non-validated questionnaires in a limited number of studies conducted in Türkiye [23,24]. Due to the necessity for an instrument that could directly ascertain HPF consumption, the Screening Questionnaire of Highly Processed Food Consumption (sQ-HPF) was developed by Martinez-Perez et al. [25], which comprises criteria derived from four [26,27,28,29] classification systems. In Türkiye, there is no validated instrument for the direct determination of HPF consumption. Within the given framework, the objective of this research is to adapt the sQ-HPF into Turkish and assess its validity and reliability.

## 2. Materials and Methods

### 2.1. Study Design and Population

At the beginning of this study, the minimum sample size was determined as 90 adults [one-way repeated measures, medium effect size (0.30), 80% power, and 5% margin of error] according to G*Power (3.0.10) analysis. Therefore, this methodological study enrolled 94 (39 male, 55 female) adults (aged 19–64 years) with the snowball sampling method who agreed to participate in this study and had blood biochemical parameter results for the last 3 months [lipid profile, fasting blood glucose, Hemoglobin A1C (HbA1C), and C-Reactive Protein (CRP)]. Pregnant or lactating women, those with an energy intake of less than 500 kcal/day or more than 3500 kcal/day for women and less than 800 kcal/day or more than 4000 kcal/day for men [30], those on a specific diet, those with severe cognitive impairment, and those with communication difficulties were excluded. The sociodemographic data of individuals were evaluated with a questionnaire form by face-to-face interview method.

### 2.2. Ethical Considerations

Ethical approval of this study was received from the Gazi University Ethics Committee (E-77082166-302.08.01-569449, dated 23 January 2023). Written informed consent was obtained from those who agreed to participate in this study by explaining the purpose of this study to the individuals in compliance with the principles of the Declaration of Helsinki.

### 2.3. Anthropometric, Body Composition, and Blood Pressure Measurements

Anthropometric measurements and body composition analysis were performed following a minimum of 4 h of fasting. Height was measured with a portable stadiometer with 0.1 cm sensitivity by the technique with the feet close together and the head in the Frankfort plane. Body weight and body composition were measured with a Bioelectrical Impedance Analysis (BIA) method using a Tanita BC730, with light clothing and without shoes. Measurement during the menstrual cycle was not performed. Waist circumference was measured with a tape measure with an accuracy of 0.1 cm. Blood pressure was measured with an automatic blood pressure monitor (Omron M6) by the researcher. The blood pressure measurement was performed from the upper arm according to the technique in [31]. The participant was instructed to remove all clothes covering the cuff, sit comfortably with legs uncrossed, back and arm supported, and upper arm cuff at right atrium level, urged to relax and not talk while measuring, and wait for 5 min before the measurement.

### 2.4. Translation of the sQ-HPF to Turkish and Cultural Adaptation

Firstly, permission from the authors of the original study to translate and validate the questionnaire into Turkish was obtained. Two researchers who are fluent in two languages translated the questionnaire into Turkish, and the food items of the sQ-HPF were modified to reflect Turkish food culture. Since the amounts of the foods in the sQ-HPF were derived from the FFQ that was previously used in the PREDIMED-Plus Cohort study [16,32], we either added quantities of foods to the Turkish version of the sQ-HPF or revised the quantities according to our market research to assure standardization and comprehensibility. After the review of the Turkish version of the questionnaire and the back-translation into English was completed, 10 expert opinions were obtained. In line with recommendations from experts, minor changes were made, and a pilot application was conducted.

### 2.5. Dietary Assessment

Three-day dietary records were utilized to evaluate daily food intake. The quantities of each food consumed were ascertained by researchers through the utilization of a photographic atlas depicting food portion sizes and quantities [33]. For the ingredients of the meals whose recipes are unknown by the participants, standard recipes in the National Menu Planning and Implementation Guide for Mass Nutrition Systems were used [34]. The Nutrition Information System (BeBis v8.2) was used to calculate the total daily intake of energy and macronutrients using the data from the 3-day dietary records. To obtain HPF % (g/day) from the 3-day dietary records, the sum of the foods (g/day) in each HPF group was calculated and subsequently divided by the total grams of the diet (g/day). To evaluate the correlation with the quantity of Additives (g/d) in 3-day dietary records, the daily salt consumption amount had to be determined. However, since it is difficult to determine the amount of added salt, it was assumed that all individuals consumed salt in the amount of 15 g/day based on the result of the SALTURK-II study conducted for the determination of salt consumption in Türkiye [35].

### 2.6. Test–Retest Reliability

For the test–retest reliability, the Turkish sQ-HPF was conducted after 4 weeks by face-to-face method with the participants who agreed to participate and provided a contact number. For the test–retest analysis, the intraclass correlation coefficient (ICC) was computed with a two-way mixed effect model and a confidence interval (CI) of 95%. ICCs were classified as follows: poor reliability below 0.5, moderate reliability between 0.50 and 0.75, good reliability between 0.75 and 0.90, and excellent reliability greater than 0.90 [36].

### 2.7. Statistical Analysis

Statistical analyses were performed using the SPSS 29.0 (IBM SPSS Statistics for Mac, Version 29.0). Data were interpreted with descriptive statistics such as mean, standard deviation, median, interquartile range (IQR), number, and percentage. Results were presented separately by gender for parameters that were considered to vary between genders. The Shapiro–Wilk test was used for testing normality. The non-parametric Kruskal Wallis test was used for the comparison of the three groups. When a significant difference was found in the Kruskal–Wallis test, pairwise Mann–Whitney tests with Bonferroni correction were performed.

The food items in the sQ-HPF were subjected to factor analysis and items with factor loadings below 0.2 were excluded from the survey as in the original questionnaire [25]. In determining whether the sample size was adequate for factor analysis, the Kaiser–Meyer Olkin (KMO) coefficient was applied, and the Bartlett test of sphericity was used to ascertain whether the items comprising the prerequisites for factor analysis were correlated. It is recommended that the KMO value should be ≥0.60 and the significance level for the Bartlett sphericity test be <0.05 [37]. Cronbach’s alpha was utilized to assess the internal consistency of the questionnaire items and a Cronbach’s alpha coefficient of ≥0.60 is considered acceptable [38].

Participants were divided into three equal sections using three tertiles according to the sQ-HPF scores, each of which represents one-third of the total. Cut-off points of the sQ-HPF scores were determined according to tertiles, and the 3rd tertile was evaluated as a high HPF consumption pattern, as in the original. The means and limits of agreement, which were defined as the mean difference ±2 times the standard deviation (SD) of the differences, were represented by horizontal lines in the Bland–Altman plot. The accepted significance level was *p* < 0.05.

For the determination of HPF consumption (% g/d) according to the Turkish sQ-HPF scores, linear regression analysis was performed. In the constant model, the Turkish sQ-HPF score was selected as a predictor and HPF (% g/d) derived from the 3-day dietary record was selected as an independent variable.

## 3. Results

Ninety-four adults (39 male, 55 female) aged 22–64 years participated in this study. The characteristics, anthropometric measurements, body fat percentage, blood biochemical parameters, and blood pressures of the participants are presented in Table 1.

To assess the representativeness of the factor (HPF diet), factor analysis was performed for the items comprising the sQ-HPF (Table 2). Three items (sugary dairy products, fermented alcohols, and distilled alcohols) were excluded from the Turkish version of the sQ-HPF since their factor loadings resulted below 0.2, similar to the original questionnaire. As previously mentioned in Section 2, to ensure standardization of the questionnaire, we added quantities to food items, in addition to the original sQ-HPF. The final version of the sQ-HPF (11 items) is presented in Table 3.

Cronbach’s alpha was found to be 0.65, which is considered acceptable/moderate, similar to the original questionnaire (0.67). The Kaiser–Meyer Olkin (KMO) coefficient for the measure of sample adequacy was found to be 0.642, which showed the sample size is adequate, and Bartlett’s test of sphericity was found significant (*p* < 0.001). Intraclass correlation coefficients (ICCs) were examined in the context of the test–retest analysis and the ICC result was 0.76 for the study population.

The scores of the Turkish version of the sQ-HPF differed from those of the original due to the removal of three items. The participants were categorized into three tertiles using a scoring system ranging from 0 to 11. Based on the result of the analysis, cut-off points for tertiles were determined as T1 → <3 points, T2 → 3–5 points, and T3 → ≥6 points. Individuals who obtained a score of 6 or higher were identified as having a high level of HPF consumption.

The correlation between HPF consumption (% g/day) obtained from the 3-day dietary records and Turkish sQ-HPF was found to be r = 0.375, *p* = 0.019; r = 0.706, *p* < 0.001; and r = 0.584, *p* < 0.001 * for males, females, and all participants, respectively.

Additionally, as a result of the linear regression analysis (*p* < 0.001), equality was obtained between the HPF consumption (% g/day) obtained from the 3-day dietary record and the Turkish sQ-HPF score as follows:HPF consumption (% g/day) = (3.465 × sQ-HPF score) + 12.354

Accordingly, the scores and the value corresponding to the HPF consumption (% g/day) are shown below:

123456789101115.819.322.726.229.733.136.640.143.547.050.5

As seen in Table 4, sQ-HPF scores, HPF % (g/day), which correspond to the score in the formulation, and HPF % (g/day), which was derived from the 3-day dietary records, significantly differed among the tertiles (*p* < 0.001). Moreover, a notable disparity in the energy derived from HPF (kcal/d) was observed among tertiles (*p* < 0.001), despite the absence of a statistically significant distinction in the total energy consumption (kcal/d) (*p* > 0.05).

While energy provided by protein decreased and carbohydrate increased among the tertiles (*p* < 0.05), the proportions of energy provided by fats did not differ in a statistically significant way (*p* > 0.05). When analyzing the daily consumption of food items in the sQ-HPF based on tertiles, although the consumption amount of other items varied according to tertiles, only a statistically significant difference was seen in the “Fats”, “Sugary and artificially sweetened drinks”, ”Refined cereals”, “Additives”, and “Fried foods” items (*p* < 0.05).

Figure 1 presents a Bland–Altman plot that demonstrates the differences and the upper and lower limits of agreement for the %HPF (g/day) obtained from the 3-day dietary records, as well as the %HPF (g/day) obtained from the sQ-HPF. The Bland–Altman plot did not exhibit an over-scattered pattern, and the majority of the values were within the range of 2SD above and below the mean (Figure 1).

## 4. Discussion

Worldwide, the consumption of HPFs is increasing [29,39,40] and there is mounting evidence of the association between HPF consumption and diet-related NCDs [41,42,43,44]. Obesity is also a contributing factor in the development of NCDs [45]. The WHO European Regional Obesity Report 2022 revealed that Türkiye has a comparatively higher prevalence of obesity compared to other European nations [46]. Thus, it is crucial for our country to accurately and reliably assess the consumption of HPFs, as it is linked to both obesity and the onset of NCDs.

In this study, we aimed to evaluate the validity and reliability of the sQ-HPF for the Turkish population. Due to the questionnaire originating in a foreign country, there were variations in the food availabilities and preferences influenced by different cultures. Therefore, we first translated the questionnaire and modified the food items to reflect Turkish food culture. However, as a result of factor analysis (Table 2), three food items were identified as insufficient to represent the study population and were excluded from the sQ-HPF due to their factor loading being below 0.2, as in the original questionnaire [25]. The internal consistency of the items in the questionnaire was evaluated using Cronbach’s alpha. Similar to the original questionnaire (0.67), the Cronbach’s alpha value was 0.65, which indicates an acceptable level of consistency [38]. Unlike the original questionnaire, our validated sQ-HPF consisted of 11 food items. Each food item was assigned a score of 1 if the participant responded to the consumption with “yes” and the scoring ranged between 0 and 11; as the score increased, it corresponded to the increment of HPF consumption.

In the literature, the studies usually assessed the consumption of highly/ultra-processed foods via dietary records, FFQs, or 24 h recalls [18,19,20,21,22]. Nevertheless, these methods for determining dietary consumption can be complex and time-consuming; therefore, an instrument that directly measures the consumption of HPFs would be consequently less challenging. Therefore, the sQ-HPF can directly provide information about HPF consumption patterns. A 3-day dietary record is the most preferred method used to evaluate food consumption [30]. Unlike 24 h recalls, the 3-day dietary record does not include the recall factor and is not as long and time-consuming as FFQs. Accordingly, we preferred 3-day dietary records to compare the sQ-HPF for the validation. The percentage corresponding to the score obtained from the sQ-HPF was estimated and equated by linear regression, and in the 3-day record, the HPF items in the sQ-HPF were proportioned to foods consumed all day (g/day). As a result, the sQ-HPF was found to be statistically correlated for both women and men (*p* < 0.05). Moreover, we assessed both total daily energy consumption (kcal/d) and energy provided by the HPFs (kcal/d). Consequently, upon dividing individuals into three tertiles based on their sQ-HPF scores, a significant distinction was observed in the energy derived from HPFs among the three groups (*p* < 0.001), even though the total energy consumption remained constant (*p* > 0.05) (Table 4). This is a contributing factor to the enhanced precision of the tool. Also, the test–retest analysis revealed that the sQ-HPF has a good reliability (ICC = 0.76).

### Strengths and Limitations

The strengths of this study include validating a concise and efficient tool for measuring HPF consumption, which is increasingly important due to the global rise in HPF consumption and its relationship with NCDs. This study fills a gap as no previous study in this context has been conducted in Türkiye. Additionally, a 3-day dietary record was utilized to enhance the validation’s strength.

However, this study has some limitations: obtaining the 3-day dietary records was challenging because this study was conducted on a voluntary basis. This prevented the sample size from reaching a high effect, resulting in a medium effect instead. The amount of salt consumption was assumed based on the results of a nationwide study in Türkiye because standard recipes specific to meals are not used uniformly among households. This variation in dishes and the difficulty in calculating the added salt at the table further complicate the assessment.

## 5. Conclusions

Consumption of HPFs is increasing worldwide and this increase is associated with the increase in diet-related NCDs. Therefore, firstly assessing and then reducing the consumption of HPFs are suitable targets for public health promotion and interventions. As a result, the Turkish version of the SQ-HPF was found to be a valid and reliable tool that is suitable for use in both epidemiological and clinical studies to evaluate HPF consumption patterns.

## Figures and Tables

**Figure 1 nutrients-16-02552-f001:**
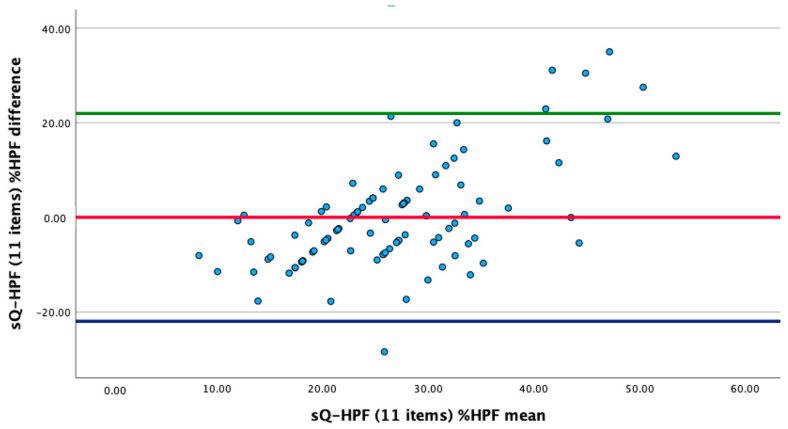
Bland–Altman graph for the %HPF (g/day) (n = 94).

**Table 1 nutrients-16-02552-t001:** General characteristics of the participants.

Variables	Male (n = 39)	Female (n = 55)	Total (n = 94)
Age (years)	51.5 ± 11.23	45.82 ± 10.91	48.2 ± 11.34
Civil status			
Single (%)	23.1	25.5	24.5
Married (%)	76.9	74.5	75.5
Education level			
Primary (%)	17.9	25.5	22.4
Secondary (%)	46.2	34.5	39.3
College (%)	35.9	40.0	38.3
Weight (kg)	86.6 ± 12.72	68.8 ± 12.45	76.2 ± 15.29
BMI (kg/m^2^)	28.4 ± 3.94	26.7 ± 4.68	27.4 ± 4.45
Waist circumference (cm)	105.0 ± 11.35	93.1 ± 12.18	98.0 ± 13.17
Body fat (%)	26.0 ± 6.00	33.68 ± 6.88	30.5 ± 7.54
Glucose (mg/dL)	106.1 ± 34.21	92.4 ± 17.24	98.1 ± 26.40
Total cholesterol (mg/dL)	187.7 ± 45.20	188.3 ± 38.72	188.1 ± 41.30
Triglycerides (mg/dL)	188.5 ± 108.28	109.7 ± 67.55	141.4 ± 94.09
LDL-C (mg/dL)	115.0 ± 41.90	110.4 ± 31.03	112.3 ± 35.73
HDL-C (mg/dL)	41.1 ± 8.63	54.2 ± 15.08	48.8 ± 14.29
HbA1C (%)	6.1 ± 1.12	5.4 ± 0.56	5.7 ± 0.91
CRP (mg/L)	11.5 ± 37.03	3.1 ± 5.58	6.7 ± 24.77
Systolic blood pressure (mmHg)	119.9 ± 17.36	105.3 ± 12.72	111.3 ± 16.41
Diastolic blood pressure (mmHg)	81.0 ± 20.40	74.2 ± 13.88	77.0 ± 17.13

*BMI:* Body Mass Index, *LDL-C:* Low-Density Lipoprotein, *HDL-C:* High-Density Lipoprotein Cholesterol, *HbA1C:* Hemoglobin A1C, and *CRP:* C-Reactive Protein.

**Table 2 nutrients-16-02552-t002:** Factor analysis for the Turkish version of the sQ-HPF.

HPF Diet	Factor Loadings
Sauces	0.808
Sugary and artificially sweetened drinks	0.695
Fried foods	0.512
Snacks	0.514
Fats	0.474
Ready-to-eat products	0.448
Refined cereals	0.414
Fatty dairy products	0.398
Cured meats	0.353
Sweets	0.311
Additives	0.233
Sugary dairy products	0.088
Distilled alcohols	−0.020
Fermented alcohols	NA

**Table 3 nutrients-16-02552-t003:** Screening Questionnaire of Highly Processed Food Consumption (sQ-HPF).

	Food Groups	Foods and Quantities	Frequency	Yes (1 Point)	No (0 Points)
1	Fatty dairy products	Cream (120 g), smoked or cured cheese (50 g), old cheddar (50 g), soft cheese (2 wedges, 25 g)	≥2 t/week		
2	Cured meats	Ham (30 g), salami/sucuk/sausage/bacon (50 g), sandwich with salami/sucuk/sausage/bacon (30 g), cured cold meat (50 g), pate (25 g)	≥1 t/day		
3	Fats	Margarine (12 g), butter (12 g), lard/animal fat (10 g)	> 3 t/month		
4	Sugary and artificially sweetened drinks	Soft drinks (200 mL, 1 small can), artificially sweetened drinks (200 mL, 1 small can), bottled juices (200 mL, 1 small can), grape must/hardaliye (100 mL)	≥2 t/week		
5	Sweets	Ice cream/sorbet (70 g), canned fruit (2 pieces), biscuits (50 g), whole grain biscuits (50 g), chocolate biscuits (50 g), honey (1 dessert spoon), molasses (1 dessert spoon), oven-baked homemade desserts (50 g), candies (50 g), donut (1 piece), cakes (muffin, cupcake, etc.) (1–2 pieces/slice), packaged cakes (50 g), sherbet desserts (100 g), chocolates (30 g), chocolate/cocoa drink powders (1 dessert spoon), hazelnut spread/cream (40 g), marzipan/flour cookies (90 g), jam (1 dessert spoon)	>1 t/day		
6	Snacks	Packaged potato crisps (50 g), packaged snacks (50 g)	>3 t/month		
7	Ready-to-eat products	Pizza (1 portion, 200 g), croquettes (1 portion), instant soup (1 bowl/plate)	>3 t/month		
8	Refined cereals	White bread (75 g, 3 slices), sliced packaged bread (75 g, 3 slices), breakfast cereals (30 g), spaghetti/pasta/noodles (dry weight 60 g), white rice (dry weight, 60 g)	≥2 t/week		
9	Sauces	Mustard (1 dessert spoon), mayonnaise (1 dessert spoon), tomato sauce/ketchup (1 dessert spoon)	>1 t/week		
10	Additives	Sugar (1 dessert spoon), table salt (1 pinch)	>3 t/day		
11	Fried foods	Eat out or homemade fried foods	≥2 t/week		

**Table 4 nutrients-16-02552-t004:** Selected dietary variables and HPF intakes according to the sQ-HPF tertiles.

Variables	T1 (<3) M (IQR)	T2 (3–5) M (IQR)	T3 (≥6) M (IQR)	*p*-Value
n	14	57	23	
sQ-HPF score	2.00 (1.25)	4.00 (2.00)	7.00 (2.00)	**<0.001 ^a^**
HPF % (sQ-HPF)	15.00 (3.70)	22.40 (7.40)	33.50 (7.40)	**<0.001 ^a^**
HPF % (3 d dietary record)	12.23 (11.00)	24.63 (10.45)	31.92 (10.45)	**<0.001 ^a^**
Energy _Total_ (kcal/d)	1312.02 (741.71)	1373.40 (678.97)	1709.27 (807.10)	0.050
Energy _HPF_ (kcal/d)	368.67 (411.60)	653.58 (531.47)	1018.14 (620.42)	**0.002 ^b^**
Protein _Total_ (%)	18.00 (5.00)	16.00 (4.00)	15.00 (4.00)	**0.047 ^c^**
Protein _HPF_ (%)	11.00 (3.00)	10.00 (4.00)	10.00 (4.00)	0.212
Fat _Total_ (%)	41.00 (7.00)	36.00 (11.00)	34.00 (10.00)	0.052
Fat _HPF_ (%)	23.00 (25.00)	26.00 (17.00)	22.00 (17.00)	0.412
Carbohydrates _Total_ (%)	39.00 (9.00)	48.00 (13.00)	49.00 (10.00)	**0.040 ^c^**
Carbohydrates _HPF_ (%)	67.00 (28.00)	64.00 (20.00)	67.00 (20.00)	0.582
Fatty dairy products (g/day)	0.00 (0.00)	0.00 (0.00)	0.00 (0.00)	0.852
Cured meats (g/day)	0.00 (0.00)	0.00 (0.00)	0.00 (5.00)	0.254
Fats (g/day)	0.00 (1.67)	1.67 (5.00)	4.17 (5.00)	**<0.001 ^c^**
Sugary and artificially sweetened drinks (g/day)	0.00 (0.00)	0.00 (66.67)	66.67 (133.33)	**<0.001 ^a^**
Sweets (g/day)	33.33 (37.21)	30.00 (85.33)	35.00 (120.67)	0.617
Snacks (g/day)	0.00 (0.00)	0.00 (0.00)	0.00 (0.00)	0.518
Ready-to-eat products (g/day)	0.00 (0.00)	0.00 (0.00)	0.00 (0.00)	0.552
Refined cereals (g/day)	69.38 (87.50)	141.67 (161.25)	198.33 (192.50)	**0.017 ^c^**
Sauces (g/day)	0.00 (0.00)	0.00 (0.00)	0.00 (0.00)	0.209
Additives (g/day)	5.00 (0.00)	5.00 (0.33)	5.00 (7.50)	**0.027 ^c^**
Fried foods (g/day)	0.00 (0.00)	0.00 (0.67)	33.33 (66.67)	**0.009 ^c^**

*T*: tertile; *M:* median; and *IQR:* interquartile range. The results are expressed as M (IQR). Bold values represent significant values based on a *p* < 0.05. ^a^ Median of all tertiles significantly differed (*p* < 0.05). ^b^ Median of T1 and T2 significantly differed to T3 (*p* < 0.05). ^c^ Median of T2 and T3 significantly differed to T1 (*p* < 0.05).

## Data Availability

Dataset available upon request from the authors due to privacy.

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
