# Peer review of "Validity and Reliability of the Turkish Version of the Screening Questionnaire of Highly Processed Food Consumption (sQ-HPF)"

_nutrients, 2024, doi:10.3390/nu16152552_

Round 1

Reviewer 1 Report

Comments and Suggestions for Authors

MANUSCRIPT: 3110174

TITLE: Validity and Reliability of the Turkish Version of the Screening Questionnaire of Highly Processed Food Consumption (sQ-HPF)

The manuscript 3110174 “Validity and Reliability of the Turkish Version of the Screening Questionnaire of Highly Processed Food Consumption (sQ-HPF)” presents an interesting work in order to prove the validity and reliability of the Turkish version of the screening questionnaire of highly processed food consumption (sQ-HPF).

This work is well structured, well planned and the research is competently carried out, the methodology was adequate for the research.

The results were subject to appropriate statistical analysis.

The literature is well cited and most of the papers cited (53.5 %) date back to the last five years.

Conclusions are not presented in a section of manuscript.

Regarding the manuscript presented, I congratulate the authors for this work of high interest to the scientific community.

However, I have some comments to be clarified and resolved in the current manuscript:

1. The manuscript presents many abbreviations without being written in full the first time they appear in the text of the manuscript (e.g. Line 40 WHO, line 65 NOVA, line 96 CRP) It is recommended to carefully review the text and present a list of abbreviations as many abbreviations are used without being defined in full the first time they are used.

2. Table 4. Please note that some results for some variables do not have superscript letters. It is recommended to carry out comparative statistical analysis to see whether or not there are significant differences in these results.

3. Table 4. Please indicate in footnote: The results are expressed as mean ± standard deviation (S.D.) and Results in same row with different letters are significantly different (p<0.05).

4. Conclusions. Please present a section with the conclusions of the work considering the results obtained. It is recommended to separate the discussion from the conclusion as some of the possible conclusions are in the discussion section.

Reviewer 2 Report

Comments and Suggestions for Authors

The manuscript describes how a food questionnaire for highly processed foods was adapted from a Spanish to a Turkish population.

The manuscript is difficult to read because it lacks explanations, is highly unstructured and results aren't clearly/correctly reported.

Specific comments:

lines 91-96: This isn't a precise description of the sample size calculation! Please provide details on all assumptions made, input data from previous studies used to inform the calculation and finally which statistical test corresponded to the sample size calculation.

lines 146-148: Please report results (such as figures) in the Results section.

Also, you don't explain how ICC was calculated. Confusing.

lines 154-155: This is not a meaningful description. Please provide much more details on the statistical methods that were used.

line 163: This is a result. Please put it in the Results section.

lines 169-173: Why are results stratified by sex? Was this declared in the Methods section?

lines 229-234: This is methodology, mostly. Please carefully separate methods and results throughout the manuscript.

lines 236-237: Again separate analyses for females and males? Why?

lines 238-241: Please describe methods such as use of linear regression and formulas in the Methods section.

Table 4: It doesn't make sense to use mean and standard deviation for summarizing right-skewed data. Please revise using median and IQR. This is a problematic table. Also, the methods used aren't described in the Methods section. Also, you need to explain what is a "Posthoc Tukey". Confusing SPSS phrase(?) in a manuscript.

Figure 1: Please the number of participants for which data are shown.

lines 272-314: Is the strengths and limitation part missing entirely? If so please include it? Please think about bias and how results generalize: a sample size of 94 is not large! And currently the sample size calculation is unconvincing.

Reviewer 3 Report

Comments and Suggestions for Authors

Dear Authors,

The manuscript (nutrients-3110174) submitted for review is quite interesting. If the manuscript were to be published, it would require refinement. The Authors' should answer reviewer's questions and do it depth revision of the manuscript. I have a few comments for the authors.

Authors, Please note and address the following comments:

Introduction

Has any research been conducted in Turkey in this area? In my opinion, the authors did not demonstrate a research gap in this area.

Material and methods

How was blood pressure measured in people participating in the study? The measurement conditions influence the result.

Is the assumption that people consumed 15 g of salt per day correct? In my opinion, no, because this is the average result for the entire Turkish population.  If this is a validation of this study, then in my opinion it should be calculated, or at least it accepted as a limitation of this study.

The methodology needs to be better described.

Conclusion

In my opinion the authors should have separated Strength and Limitation section, as well as Conclusion. Now this is a common section with discussion of the results.

References: References are not cited according to journal rules

Technical notes: The manuscript should be corrected from the technical point of view.

This manuscript requires deep correction and clarification of my comments. In my opinion, it is not suitable for publication in this form.

 Reviewer
